# The Brilliance of *Borrelia:* Mechanisms of Host Immune Evasion by Lyme Disease-Causing Spirochetes

**DOI:** 10.3390/pathogens10030281

**Published:** 2021-03-02

**Authors:** Cassidy Anderson, Catherine A. Brissette

**Affiliations:** Department of Biomedical Sciences, University of North Dakota, Grand Forks, ND 58202, USA; cassidy.anderson@und.edu

**Keywords:** Lyme disease, *Borrelia*, immune response, innate, adaptive, complement

## Abstract

Lyme disease (LD) has become the most common vector-borne illness in the northern hemisphere. The causative agent, *Borrelia burgdorferi* sensu lato, is capable of establishing a persistent infection within the host. This is despite the activation of both the innate and adaptive immune responses. *B. burgdorferi* utilizes several immune evasion tactics ranging from the regulation of surface proteins, tick saliva, antimicrobial peptide resistance, and the disabling of the germinal center. This review aims to cover the various methods by which *B. burgdorferi* evades detection and destruction by the host immune response, examining both the innate and adaptive responses. By understanding the methods employed by *B. burgdorferi* to evade the host immune response, we gain a deeper knowledge of *B. burgdorferi* pathogenesis and Lyme disease, and gain insight into how to create novel, effective treatments.

## 1. Introduction

Since the first investigations conducted by Steere and Malawista in 1975 [1,2], Lyme borreliosis, otherwise known as Lyme disease (LD), has become the most common vector-borne illness in the northern hemisphere [3]. LD is caused by infection with a member of the *Borrelia burgdorferi* sensu lato (s.l.) complex. Within the complex, three species cause the majority of LD in humans: *Borrelia burgdorferi* sensu stricto (s.s.), *Borrelia afzelii*, and *Borrelia garinii* [4]. Lyme borrelia are transferred to the vertebrate host by *Ixodes* ticks. In the northeastern and upper midwest of the US, the main vector is *Ixodes scapularis*, while *Ixodes pacificus* is the primary vector in western US [5]. *Ixodes ricinus* and *Ixodes persulcatus* are the tick vectors for the *Borrelia burgdorferi* sensu lato (s.l.) complex in Europe and Asia, respectively [5].

The most common clinical sign of Lyme disease in the United States is the formation of an erythema migrans skin lesion, which is often accompanied by flu-like symptoms. However, Lyme spirochetes are capable of disseminating to other tissues and causing other manifestations, such as Lyme neuroborreliosis, Lyme carditis, or Lyme arthritis [5]. Symptoms of LD can persist following treatment with antibiotics, resulting in a condition known as post-treatment Lyme disease syndrome (PTLDS). PTLDS is often functionally disabling and leaves patients with fatigue, cognitive complaints, or musculoskeletal pain [3].

For *B. burgdorferi* to persist in the host, the pathogen must employ a variety of tactics to evade the immune response. This review aims to provide a current overview of a majority of the immune evasion tactics employed by *B. burgdorferi*.

## 2. Innate Response

### 2.1. Complement Cascade

The first line of defense implemented by the immune system to protect the host against pathogens is known as the complement system. The complement system is a tightly regulated cascade of enzymatic proteins responsible for the opsonization of pathogens, phagocytosis, cell lysis, and the establishment of the membrane attack complex (MAC) [6,7]. There are three main activation pathways: the classical pathway (CP), the alternative pathway (AP), and the lectin pathway (LP) (Figure 1). All pathways converge at the complement protein C3 and the formation of activation products C3a, C3b, C5a, and the MAC (C5b-9) [6]. 

#### 2.1.1. Direct Interference 

*B. burgdorferi* evades the complement system in two main methods. The first method involves a direct inference with the components of the complement cascade pathways. Direct interference is accomplished through the use of outer surface proteins, the most notable of which include BBK32, OspA, OspC, and BBA70.

BBK32 is a surface lipoprotein found on Lyme spirochetes that acts as a vascular adhesin and binds both glycosaminoglycans and fibronectin [8,9]. A study by Garcia et al. found that BBK32 is capable of inhibiting the complement CP through high-affinity binding of the C1r subunit of C1 (Figure 2). After C1q binds to the surface of the *Borrelia*, BBK32 recognizes and binds to C1r, blocking the autocatalysis of the C1r proenzymes and the proteolysis of the C1s proenzymes. Since the C1 complex is the initiating step of the CP, the binding of BBK32 to C1r leaves the pathway in an inactive state. BBK32 knockout mutants were found to have reduced infectivity, indicating that mutants could have a deficiency in either adhesion or complement evasion [10]. 

The enzyme plasmin is a known inhibitor of the complement system due to its binding and cleaving of Cb3 and C5 [11]. The outer surface protein OspA has been shown to bind plasminogen; however, its expression is downregulated once the spirochete enters the vertebrate host [12]. OspC, another outer surface protein found in Lyme spirochetes, has been shown to bind to plasminogen and its expression is upregulated in the vertebrate host [13]. OspC has also been shown to bind C4b in vivo in both *B. burgdorferi* and *B. garinii* [14]. The protein BBA70 is capable of binding plasminogen with a high affinity as well, though it is not able to bind other complement regulators [15]. 

The complement-regulator acquiring surface protein (CRASP) CspA has also been shown to bind plasminogen [16]. In addition, CspA is capable of interacting with C7, C8, C9, and the MAC [17,18]. By binding C7 and C9, CspA interferes with the MAC formation through the inhibition of C9 polymerization [17,19,20]. The expression of CspA is upregulated during the tick blood meal, but downregulated in the vertebrate host, indicating that it is important in avoiding the immune response during transmission, but is not essential for continued infection [21]. Other CRASP genes *cspZ*, *erpA*, *erpC*, and *erpP* express proteins that will bind plasminogen [22]. Another species of *Borrelia*, *B. bavariensis*, expresses surface proteins BGA66 and BGA71 that share a similarity in sequence with *cspA* and are capable of inhibiting MAC. BGA66 is capable of inhibiting all three complement pathways, while BGA71 inhibits AP and CP [23]. 

Pausa et al. found a human-like CD59 protein on the surface of *B. burgdorferi*, which binds C9 and to a lesser extent the beta subunit of C8 and therefore inhibits the MAC [24]. Currently, there has been only one report on the function of CD59, and further assessment of the function and overall role of this protein in the evasion of the immune response is needed. 

#### 2.1.2. Binding of Regulators

In addition to direct interference with the complement system, CRASPs are capable of binding regulators of the complement system. Factor H (FH) is one of the main regulators of the AP (Figure 2). The bunding of FH to C3b accelerates the decay of the AP C3 convertase and promotes Factor I (FI) mediated C3b cleavage [25].

The CRASP protein CspA is essential for the survival of Lyme spirochetes. However, it is expressed by spirochetes only in feeding ticks or at the site of feeding, and not in disseminated spirochetes [26,27]. CspZ is upregulated in the vertebrate host, though it is not essential for the acquisition of spirochetes to the mammalian host [28,29]. When exposed to the spirochetal outer surface, the proteins CspA and CspZ bind complement regulators FH and factor H-like protein-1 (FHL1) [26]. Introducing CspA or CspZ to a serum sensitive strain of spirochete allows survival in vitro in various vertebrate serums, demonstrating the importance of complement evasion [29].

The outer surface protein OspE is expressed throughout the different stages of infection [30,31,32,33]. OspE binds to FH as well as different isotopes of complement factor H-related (CFHR) proteins [33,34]. The complement regulatory domain is on the N terminus, while the C terminal region contains binding sites for C3b, heparin, and microbial surface proteins and is central for FH function, target recognition, discrimination between self and non-self, and anti-inflammatory activities [34]. 

The OspE-related proteins ErpA and ErpP bind the complement factor H-related proteins CFHR1, CFHR2, and CFHR5, while ErpC binds only CFHR1 and CFHR2 [26]. Studies have shown that in the absence of FH and in the presence of CFHR1, CFHR2, and CFHR5 the complement attack is not inhibited, indicating that FH works in conjunction with CFHR1, CFHR2, and CFHR to support complement evasion [35]. However, the overall role that Erps play in immune evasion or the importance of CFHR binding still remains unclear [29]. 

#### 2.1.3. Tick Salivary Proteins 

The proteins found in tick saliva have been shown to inhibit the complement system (Figure 2). The tick salivary protein, Salp15, binds to OspC both in vitro and in vivo to protect Lyme spirochetes from antibody-mediated killing [36,37,38,39]. This is accomplished by Salp15 preventing the formation of the MAC through the inhibition of the deposition of terminal C5b-9 [40]. The expression of Salp15 was selectively enhanced in the salivary glands during transmission [41]. 

Salp15 homologs have been identified in other *Ixodes* species that are known vectors for Lyme disease [40,41]. Salp15 from *I. persulcatus* is capable of binding to OspC to protect spirochetes from antibody-mediated killing and phagocytosis. The Salp15 found in *I. ricinus* offers significantly stronger protection from complement-mediated killing from human serum compared with *I. scapularis* [41]. 

*I. scapularis* anti-complement protein (Isac), Salp20, and *I. ricinus* anti-complement proteins I and II are all part of a homologous protein family called the Isac-like protein (ILP) family, that work to inhibit the complement AP through the dissociation of the C3 convertase components, C3b, and cleaved factor B [42]. 

Salp20 inhibits the AP by binding and dissociating C3BbP, the active C3 convertase [43,44]. Inhibition of the complement system by Salp20 in murine models was accomplished at concentrations as low as 5 μg [44]; this is significant as the upper limit for any given protein in tick saliva is estimated to be 10 μg [45]. Salp20 prevents the cleavage of C3 into C3a and C3b, thus preventing the deposition of C3b to pathogen surfaces for opsonization and dissociates factor B (fB) from the covalently bound C3b, disrupting the C3 convertase [44]. This method of complement inhibition is also accomplished by *I. scapularis* anti-complement protein (Isac), as well as the closely related *I. ricinus* proteins IRAC I, IRAC II, and IXAC-B1-5 [44,46]. 

The protein properdin (factor P) binds and stabilizes the complement factor C3bBb and suppresses the activity of FH [47]. Properdin that has bound C3b inhibits the FH cofactor ability, impeding the FH-dependent decay acceleration of C3bBb [47,48,49]. Salp20 displaces properdin from C3bP, which leaves C3 vulnerable to FH-mediated cleavage. Salp20 can also displace properdin from C3bBbP, leaving C3bBb vulnerable to FH-mediated decay [44,49]. 

The Tick Salivary Lectin Pathway Inhibitor (TSLPI) is a dominant complement inhibitor in tick saliva [50]. TSLPI reduces complement-mediated killing and interferes with the complement LP cascade by interfering with the mannose-binding lectin (MBL)-dependent C4 activation [51]. This interference results in impaired neutrophil phagocytosis and chemotaxis and diminished lysis of *Borrelia* [52]. 

The infection of *I. scapularis* nymphs with *B. burgdorferi* s.s. resulted in a higher expression level of TSLPI mRNA after 24 h of tick attachment compared to uninfected ticks [51]. In *I. ricinus* ticks, an ortholog of TSLPI is upregulated during tick feeding but was not present in unfed ticks, similar to the ortholog found in *I. scapularis* ticks [50,51]. 

### 2.2. Antimicrobial Protein and Peptide Resistance

The host innate immune system produces antimicrobial proteins and peptides in response to pathogens. *B. burgdorferi* has demonstrated resistance to antimicrobial proteins lactoferrin, azurocidin, and proteinase 3, as well as a limited susceptibility to lysosomes [53]. The resistance to lactoferrin, an iron-binding and transport protein, is due, in part, to the fact that *B. burgdorferi* does not require iron [54].

*B. burgdorferi* is also highly resistant to the antimicrobial peptide cathelicidin [55]. The antimicrobial resistance of *B. burgdorferi* is thought to be due to the lack of lipopolysaccharide (LPS) in the outer membrane. LPS are typically found in Gram-negative bacteria, where cationic peptides, such as cathelicidins, can bind to the molecule [55,56]. Salp15 also works to inhibit cathelicidin, as well as human defensins (hBD-2 and hBD-3), and psoriasin [56]. 

In addition to resistance to antimicrobial peptides, *B. burgdorferi* expresses the surface protein BBA57, which was found to decrease the transcription of antimicrobial peptides. A recent study found that BBA57 decreased the expression of antimicrobial peptides (AMP), bactericidal/permeability-increasing protein (*Bpi*), lactotransferrin, and secretory leukocyte proteinase inhibitor (*Slpi*). This protein is conserved within the *B. burgdorferi* s.l. and lacks homology with other proteins of known function. BBA57 was found to be critical for the early stages of infection, but not for later-stage persistence. The exact mechanism of the suppression of AMP is unknown, but it is thought to be mediated by OspC [57].

### 2.3. Macrophage Interference 

*Borrelia* can cause an increase in the production of IL-10, an anti-inflammatory interleukin [58,59,60]. Macrophages and dendritic cells are major producers of IL-10 and are known to downregulate immune mechanisms in the presence of IL-10 [58,61]. IL-10 suppresses the secretion of proinflammatory cytokines TNFα, IL-6, and IL-12 produced by macrophages and dendritic cells in the presence of *B. burgdorferi* [58]. IL-10 production also leads to suppression of phagocytosis by macrophages and a decrease in the production of proinflammatory mediators and co-stimulatory molecules in antigen-presenting cells (APCs) [58]. Studies performed on IL-10^-/-^ mice found that a lack of IL-10 resulted in a 10-fold greater clearance of *B. burgdorferi*, indicating the importance of IL-10 production to the evasion of immune clearance [59]. 

Macrophages and dendric cells act as APCs and activate the T cell response to pathogens through antigen presentation on MHC II molecules, providing a bridge between innate and adaptive immunity [58,62]. During *Borrelia* infection, dendritic cells downregulate co-stimulatory receptors, except for CD86, which was found to be upregulated. It is thought that the reduced response of co-stimulatory receptors is due to IL-10 [58]. 

### 2.4. Disabling of Chemokines and Alarmin Molecules

In addition to disabling the complement system, tick saliva also contains a chemokine-inhibitory evasin protein. Chemokine proteins are responsible for the recruitment of leukocytes to the site of infection, where they bind to the chemokine receptors and direct leukocytes to the site of the tick bite [63]. Evasins are widely expressed within the Ixodidae family, are found in tick saliva, and reduce the migration of immune cells to the site of infection by inhibiting the recruitment of leukocytes [63]. Evasins inhibit the binding of chemokines to glycosaminoglycans, inhibiting chemokine activity [63,64]. Tick saliva proteins inhibit chemokines CCL2, CCL3, CCL5, and CCL11 [64]. (For a comprehensive review of evasin classification and activity, see Bhusal et al. 2020 [65].

In addition to their antimicrobial, enzymatic, or chromatin-binding functions, AMPs can act as “alarmin molecules” to initiate migration and activation of APCs [66]. Specific motifs from conserved bacterial structures, known as pathogen-associated molecular patterns (PAMPs), stimulate antigen-presenting cells (APCs) through Toll-like receptors (TLR) [67,68]. Activation of TLRs results in the differential expression of chemokines and cytokines [67,68]. 

Salp15 was found to inhibit the mRNA expression of chemokines, monocyte chemoattractant protein 1 (MCP-1), and IL-8, as well as the expression of the alarmin molecules hBD-2, hBD-3, RNase 7, and psoriasin [56]. The inhibition of chemokines and alarmin molecules and the subsequent migration of leukocytes and APCs to the site of infection allows for *B. burgdorferi* transmission, multiplication, and dissemination. 

### 2.5. Neutralize Reactive Oxygen Species

At the site of infection, neutrophils release what is known as an “oxidative burst” of reactive oxygen species (ROS) to combat infection. The production of ROS and reactive nitrogen species (RNS) is essential to the destruction of bacterial pathogens. ROS consist of superoxide radicals (O^−^_2_), hydrogen peroxide (H_2_O_2_), and hydroxyl radicals (OH^−^), and RNS consists of nitric oxide (NO), dinitrogen trioxide (N_2_O_3_), nitrogen dioxide (NO_2_), and peroxynitrite (NO_3_^−^) ROS can oxidize cysteinyl residues, iron–sulfur clusters, DNA, polyunsaturated lipids, proteins, and cellular membranes [69]. 

ROS are most damaging due to the Fenton reaction, where H_2_O_2_ and Fe^2+^ interact to produce OH. Fe^2+^ is found along the phosphodiester backbone of DNA, and the OH produced from the Fenton reaction can react with deoxyribose and damage DNA [69]. *B. burgdorferi* was found to be resistant to ROS-mediated DNA damage when compared to other bacterial pathogens. *B. burgdorferi* is iron independent, encoding few genes of known iron-containing proteins, and does not require iron for growth [54]. A study by Chung et al. found that IL-10, which is increased in production during *B. burgdorferi* infection, significantly reduces the production of both ROS and NO by macrophages, providing further protection for the pathogen [58]. 

The lipid membrane of *B. burgdorferi* was found to be a target of damage by ROS. Free radicals attack the polyunsaturated fatty acids in the cell membranes and initiate lipid peroxidation. This results in a decrease in membrane fluidity, altering the physical properties of the membrane. Oxidation only occurs with certain lipids, such as linoleic and linolenic acid, which *B. burgdorferi* scavenges from its surroundings and incorporates into its membrane. Around 10% of the total lipid concentration in *B. burgdorferi* membranes was found to be made up of linoleic acid, while linolenic acid had a concentration of around 1% of the membrane [70]. 

In order to mitigate cellular damage caused by ROS, bacteria require antioxidant defenses. One of the most important are enzymes known as superoxide dismutases (SODs) that are used to break down superoxides. *Borrelia* utilizes the manganese-dependent (Mn) SOD SodA, an essential virulence factor with clear contributions to mammalian infection [71]. It is unclear if the role of SodA is to detoxify ROS produced by the innate immune response or endogenous ROS produced by the organism itself. High levels of manganese are required to activate SodA. The limitation of the bioavailability of manganese by macrophages and neutrophils to, *B. burgdorferi* is also ineffective, as *B. burgdorferi* are still capable of accumulating enough manganese to activate SodA [72]. The metal transporter A (BmtA) is responsible for *B. burgdorferi’s* uptake of manganese and is essential for the infection of mammals [73]. *B. burgdorferi* Mn-SOD protects intracellular targets and not the membrane from ROS damage [54]. 

While *B. burgdorferi* is more resistant to ROS and RNSs than other pathogenic bacteria, it is still susceptible to damage by H_2_O_2_ [70].

### 2.6. Pleomorphic Forms

*B. burgdorferi* s.l. is a pleomorphic bacterium, and therefore is capable of changing its morphology based upon varying environmental conditions. Beyond the most common spirochete conformation, *B. burgdorferi* can exist as round bodies (RBs) or in a biofilm-like (BFL) aggregation [74]. 

*B. burgdorferi* have been seen to change conformation from spirochetes to the spherical RBs during harsh conditions in vitro [74]. In vivo observations of RBs have been noted in few clinical studies, as well [75,76]. A study by Meriläinen et al., found that *Borrelia* which had entered the RBs conformation were capable of reverting back to viable and motile spirochetes. RBs *Borrelia* were found to have reduced levels of metabolic activity, though reverted spirochetes were noted to have normal metabolic levels [74]. 

RBs formation has been thought to enhance the survival of the bacteria in poor environmental conditions and the evasion of the immune system [77,78,79,80]. The lower metabolic activity of RBs may aid in the survival of the bacteria during antibiotic treatments, though RBs could only withstand exposure to harsh environments for short spans of time [74]. *B. burgdorferi* may have additional strategies to evade antibiotic treatment, including antibiotic tolerance, although the mechanisms remain undefined [80,81,82,83,84].

Biofilms are a complex aggregate of microorganisms that bacteria and other microorganisms use to protect themselves from the hostile host environment [85]. In response to the extreme environment, the bacteria secrete extracellular polymeric substances (EPS) to act as a shield against stressors [85,86]. Biofilm production may help *Borrelia* to survive in extreme environmental conditions, such as non-physiologic pH, extreme temperature, high concentration of metals, the addition of xenobiotics, or antimicrobials [86,87,88]. It has been hypothesized that the aggregation of *B. burgdorferi* may aid in the binding of the bacteria to host tissues and may allow the bacteria to avoid phagocytosis [89]. 

*B. burgdorferi* biofilm-like growth in suspension and on surfaces been observed in many studies [88,89,90]. Though the spirochete conformation is the most commonly observed, BFL aggregations are observed at relatively low concentrations [74]. Meriläinen et al. found that BFL colonies were a part of *B. burgdorferi*’s normal in vitro growth and that the colonies formed before the bacteria reached the exponential growth phase [74]. The aggregation of *Borrelia* occurs preferentially in conditions with high temperatures, low pH, and high cell density in vitro [89]. 

*Borrelia* aggregates are comprised of extracellular polysaccharides, similar to that of other microorganism biofilms [85,86]. *Borrelia* biofilms also have channels, similar to that produced by *Leptospira* spp., that provide oxygen and nutrients to the aggregates, as well as the removal of waste [86].

Sapi et al. 2016 was the first study to show the presence of a *Borrelia* biofilm in human skin tissue. The study found that *B. burgdorferi* s.s. and s.l. have specific surface biofilm markers, like alginate, a biofilm marker found in other pathogenic bacteria [90]. The extracellular polysaccharides found in biofilms play an essential role in the protection of pathogens, immune evasion, and antibiotic resistance [90]. *Borrelia* exhibit a preference for collagen and fibronectin surfaces for biofilm production [90]. 

Suspension biofilms found in normal cultures were found by Meriläinen et al. to produce proteins in an extracellular polymeric substance (EPS) matrix, the most notable of which was collagen. It was reasoned that the presence of collagen in the EPS may promote binding of the suspended biofilm to the host tissue [74].

However, the presence of *B. burgdorferi* biofilms or pleomorphic forms in vivo is controversial. There is still much more research that needs to be performed in order to elucidate the potential roles of RBs and biofilm formation in *B. burgdorferi* persistence and the evasion of the immune system. 

### 2.7. Intracellular Localization

*Borrelia burgdorferi* is capable of hiding from the immune system in vitro by invaginating itself through binding to fibrocytic cells [91]. *Borrelia* can also enter endothelial cells and macrophages. While intracellular localization of *B. burgdorferi* appears to be a rare occurrence, the ability of *Borrelia* to be internalized and survive within host macrophages could represent a potential reservoir for chronic or reoccurring Lyme disease [92]. 

Studies have shown that *Borrelia* spirochetes are capable of intracellular localization. It was found that actin-containing microfilaments were required for intracellular localization and that the host cell is a participant in the process [93].

A study conducted by Wu et al. 2011 found that intracellular localization of *B. burgdorferi* into mammalian cells led to brief protection from antibiotic killing. *Borrelia* were shown to invade several non-phagocytic cells, such as endothelial cells, fibroblasts, neuronal and neuroglial cells. *B. burgdorferi* that did not synthesize the β1 integrin subunit had a reduced capability of invading fibroblasts, indicating a necessity for β1 integrin for borrelial invasion [94]. The ability for *B. burgdorferi* to form long-term co-cultures with primary human fibroblasts supports the concept of intracellular localization aiding *Borrelia* in immune evasion. 

## 3. Adaptive Response

### 3.1. The Humoral Response

The humoral response is one of the main mechanisms by which the adaptive immune response operates. The humoral response works to protect the extracellular spaces of the host, through the production of antigens from activated B cells in the lymph nodes [95]. During early infection, *Borrelia* target the lymph nodes where they continue to occupy the lymphoid tissue for the duration of infection. Once there, *B. burgdorferi* causes rapid B cell proliferation, leading to the enlargement of the lymph nodes, structural damage, and the deterioration of the T and B cell zones in mice [96,97,98]. 

B cells are the primary adaptive response for the clearing of *B. burgdorferi* infections in mice, as B cell deficiencies were found to lead to a more severe illness [99]. The increased cell accumulation in the lymph nodes was due to CD19+ B cells, of which some produced antibodies specific for *B. burgdorferi*. There was a lack of CD4+ T cell accumulation in the lymph nodes [97]. A study by Elsner et al. showed that despite their induction, the CD4+ T cells and T follicular helper cells did not function properly and led to rapid B cell proliferation but not differentiation in vivo [100]. The B cells packed in the lymph nodes lacked the typical follicular arrangement and the CD4+ T helper cells were scattered and did not preside inside discernable T cell zones [97].

A study by Hastey et al. indicates that *B. burgdorferi* infection induces type I IFN signaling that increases the proliferation of B cells in draining lymph nodes and aids in the disruption of the lymph node architecture [101]. 

The invasion of the lymphoid tissue disrupts the ability of the immune response to form functioning germinal centers (GC). GC form in the secondary lymphoid tissue during infection and are required for the formation of long-lived plasma cells to aid the immune response by continuously secreting antibodies and inducing memory B cells [102]. 

During infection with *B. burgdorferi*, GC formed in the first two weeks of infection, but they were short-lived and rapidly dissipated over the following two weeks [96,97]. The GC demonstrated changes in structure and was incapable of inducing memory B cells and long-living plasma cells for several months post-infection. Mice that were co-administered a vaccine at the time of infection with *B. burgdorferi* failed to produce antibodies to the vaccine antigen. The temporary immunosuppression leaves the host open for reinfection with the same strain of *B. burgdorferi*, especially if the infection was treated with antibiotics. This is also seen in endemic areas, where reinfection with Lyme disease is common [96]. 

Tick saliva also inhibits the production of antibodies by plasma cells, though this occurs only at the site of infection and has no effect on the formation of memory B cells [103,104]. Salp15 from *I. scapularis* was found to have an immunomodulatory effect through the inhibition of CD4+ T cell activation and the production of IL-2 in a dose-dependent manner in vivo [105]. The binding of Salp15 to CD4+ T cells is persistent and produces a long-lasting immunomodulatory effect, resulting in a reduction in the production of cross-antigenic antibodies [106]. 

T follicular helper cells (THD) and follicular dendritic cells (FDC) are also crucial to the function of GC [107]. TFH cells were found to rise in numbers in the GC during *B. burgdorferi* infection, but rapidly declined to pre-infection numbers after 45 days. The FDC of some GC failed to correctly position themselves opposite of the T cell zone. This disruption of the FDC network could inhibit the ability of the GC to properly function [96].

The presence of *Borrelia* may result in a lower deposition of the complement component C4 onto FDC. As mentioned earlier, the protein BBK32 on the surface of the spirochete inhibits the classical pathway and prevents the formation of C4. It is possible that the lack of antigen-C4 deposition reduces the antigen presentation by the FDC to the GC B cells, which would cause a premature collapse of the GC [96].

*B. burgdorferi* induces a T cell-independent B cell response [99]. A study by Elsner et al. found that there was a failure to produce long-lived antibodies to T cell-dependent Borrelial antigens. T cell-independent antigens produce short-lived antibodies that last only as long as the infection, while T cell-dependent antibodies, which are generated by the GC, produce a long-lived response [100]. 

*Borrelia* infection results in a failure of the B cells to undergo a class switch recombination of IgM to IgG antibodies [97]. Several studies have shown that the number of IgM-secreting cells exceeds that of IgG-secreting cells in both mouse models and human patients [96,108,109]. The B cell response produces unusually strong and persistent IgM antibodies that are both T cell dependent and T cell independent. The failure to produce a strong T cell-dependent response or long-lived germinal centers leads to a humoral response that is dominated by IgM-secreting B cells in both the lymph nodes and the bone marrow [97]. IgG-secreting cells did accumulate slowly in the bone marrow, but it was insufficient to clear the infection, and production ceased around the same time as the collapse of the GC [97,100]. The animal data are bolstered by clinical evidence for elevated and long-lasting IgM responses in human patients with persistent symptoms and/or late manifestations of Lyme borreliosis [109,110,111,112,113].

IgM may work to clear *Borrelia* from the blood, but due to its large size, it could fail to reach and clear an infection in the skin. The large and sustained IgM production is indicative of a failure of the B cells to undergo a class switch recombination to an IgG antibody production. This could affect the ability of the host immune response to clear infection in tissues unreachable by IgM antibodies. It is thought to be most likely that the cause for the high IgM production is a failure of the B cell response to undergo a class switch due to interference from *B. burgdorferi* [97]. 

### 3.2. Antigenic Variation

Antigenic variation in Lyme *Borrelia* is an extensively researched immune evasion tactic. This section aims to provide only a brief overview of the subject; for a more thorough review, please refer to [114,115,116].

Antigenic variation (Figure 3) is a common evasion tactic employed by pathogens such as bacteria, protozoans, and fungi. While the host’s adaptive immune response works to produce the antigen-specific antibodies to clear infections, the pathogen has created a new variation of the antigen and is now unrecognizable by the antibodies being produced by the host. This variation of the surface antigen is generated by recombination events that produce altered versions of the proteins, by changes in the allele expression, or both [115]. 

In Lyme *Borrelia*, the *vls* locus is the site of recombination for antigenic variation. The *vls* locus is located on linear plasmid 28-1 (lp28-1) for the *B. burgdorferi* strain B31. The expression locus, *vlsE*, encodes for the outer surface lipoprotein. The *vlsE* gene is located near a hairpin telomere and has 15 silent cassettes located adjacent to and upstream of *vlsE* going in the opposite orientation [116]. The expressed protein, VlsE, is a surface-bound lipoprotein that is continuously modified as the gene goes through segmental gene conversion events with the silent cassettes [114]. Please see Chaconas et al. 2020, Figure 2, for a detailed visual of the *vls* locus and recombination process [114].

The locus is the most evolutionary diverse gene in the pathogen [114]. Lyme *Borrelia* genomes typically show a high degree of conservation, for example, the RecA protein from nine different *Borrelia* species show 95% sequence identity. However, comparisons of *VlsE* in *B. burgdorferi* strains B31 and 297 showed only 46% identity, while B31 and three other Lyme species share only 35–49% [116].

The *vlsE* gene is essential for the initial and persistent infection of Lyme, as the absence of lp28-1 resulted in low infectivity of *B. burgdorferi* [117,118,119]. The variation of *vlsE* has also been found to be required for reinfection and is an advantage for the enzootic cycle [120,121,122]. Three-dimensional modeling was used to determine that amino acid changes that resulting from recombination events with the silent cassettes to the *vlsE* are exposed and accessible to antibodies [123]. 

B31 has an inverted repeat (IR) upstream of *vlsE*, in the prompter region, that is 100 bp in length [123]. Under conditions that cause negative supercoiling, such as replication or transcription, the IR can take on a cruciform structure [114,124]. In this event, the cruciform could act as a flag for recognition of proteins involved in the recombination events, as it is a distinct marker not found in the silent cassettes [114]. There are proteins that use cruciform as recognition sites for replication, recombination, and repair. In addition, cruciform can be sites for the introduction of double or single-stranded breaks [125,126]. The exact role of the IR has yet to be demonstrated, but it is thought that it could play a role in *vlsE* transcription or recombination [114]. 

The locus has several conserved regions, despite its ability to constantly change the genetic code at this location. These conserved regions include the telomeric location of the locus, the inverse orientation of *vlsE* and the silent cassettes, the inverted repeat consisting of 100 bp near the 5′ end of *vlsE*, and the high concentrations of G runs in *vlsE* and the silent cassettes [114]. There may also be well conserved direct repeats (DR) present that are 17 bp in length and flank the cassettes and variable region of *vlsE* [114,126]. 

The DR found in B31 contains G-runs that form intermolecular G-quadruplex structures in vitro, though it is not known whether the G-quadruplex can be formed in vivo. A G-quadruplex, also known as G4 or 4-stranded DNA, is stabilized by Hoogsteen hydrogen bonding of the bases. The G-quadruplex could play a role in the gene conversion events that occur at the *vlsE*. The *vls* locus is rich in G-C nucleotides, tacking up 48% of the locus. This is in comparison to a very A-T-rich genome, with only 29.75% of the genome comprised of G-C pairs. The frequency of G-runs is also much higher than to be expected [116]. This was observed in several different *B. burgdorferi* strains. The G-runs are not frequently found on either the coding or non-coding strands of the non-*vls* DNA on the plasmid that carries the *vls* locus. The G-runs are found in high numbers on the coding strand. It is believed that there is an essential function that the G-C-rich content and the G-runs serve due to its preservation [114]. 

G4 DNA is an inhibitor of DNA replication and can show specificity for either the leading or the lagging strand [127,128]. G4 may facilitate interactions and synapsis from distant locations, or the non-specific formation of G4 could provide sites of stalled DNA replication, where recombination is prevalent [129,130,131]. 

The mechanism for the recombination events is still largely unknown, but it has been shown that the RuvAB Holliday junction branch migrase is required. The cis location of the *vlsE* and the silent cassettes, as well as the high G-C content and the GC skew, may be another requirement for recombination [116]. 

## 4. Conclusions

*B. burgdorferi* utilizes several different methods to evade the host immune response. *Borrelia* disable the complement system through the regulation of outer surface proteins, the binding of complement regulators, and the use of tick salivary proteins. The innate immune response signaling through chemokines and alarmin molecules is disrupted through the use of tick salivary proteins, preventing the migration of immune cells to the site of infection, allowing *B. burgdorferi* to establish infection. Resistance to antimicrobial proteins and ROS-mediated killing, as well as the disabling of macrophages, prevents the removal of *B. burgdorferi* spirochetes from the host. Pleomorphism and intracellular localization may also play a role. 

The adaptive immune response is disabled through the invasion of the lymph nodes by *Borrelia*, and the resulting collapse of the germinal center, the lack of memory cell production, and antibody class switching. *B. burgdorferi* also evades detection through the antigenic variation of the outer surface protein VlsE. 

Though research in recent decades has shed light on the mechanisms by which *B. burgdorferi* evades the immune response, there is still much to be learned. More research is needed to elucidate the exact mechanisms that *B. burgdorferi* uses to interact with and evade the different parts of the immune response. A better understanding of how *B. burgdorferi* subverts the host immune response is important to the development of novel treatments and preventative measures. 

## Figures and Tables

**Figure 1 pathogens-10-00281-f001:**
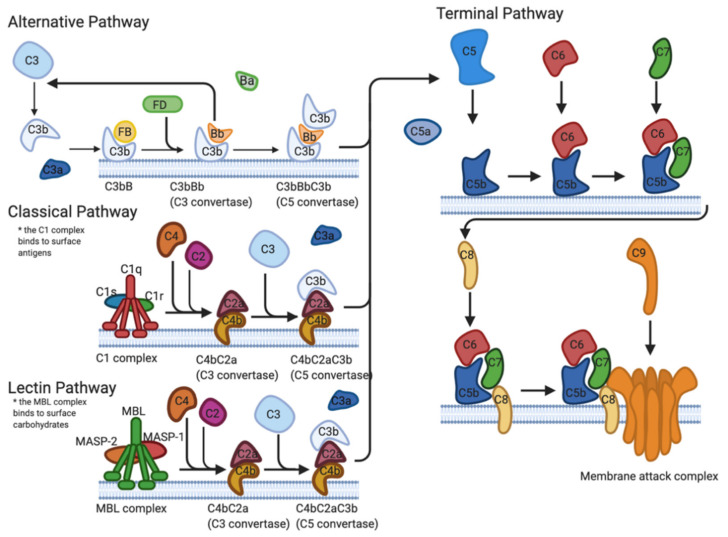
Formation of the membrane attack complex formation via the alternative, classical, and lectin pathways. The alternative pathway is activated by the binding of C3b directly to the surface of a microbe. The classical pathway activation is triggered by the presence of antigen–antibody complexes. The lectin pathway is activated by the binding of mannose-binding lectins to mannose-containing liposaccharides on the surface of the pathogen. Created with BioRender.com; https://biorender.com/ (accessed on 1 March 2021).

**Figure 2 pathogens-10-00281-f002:**
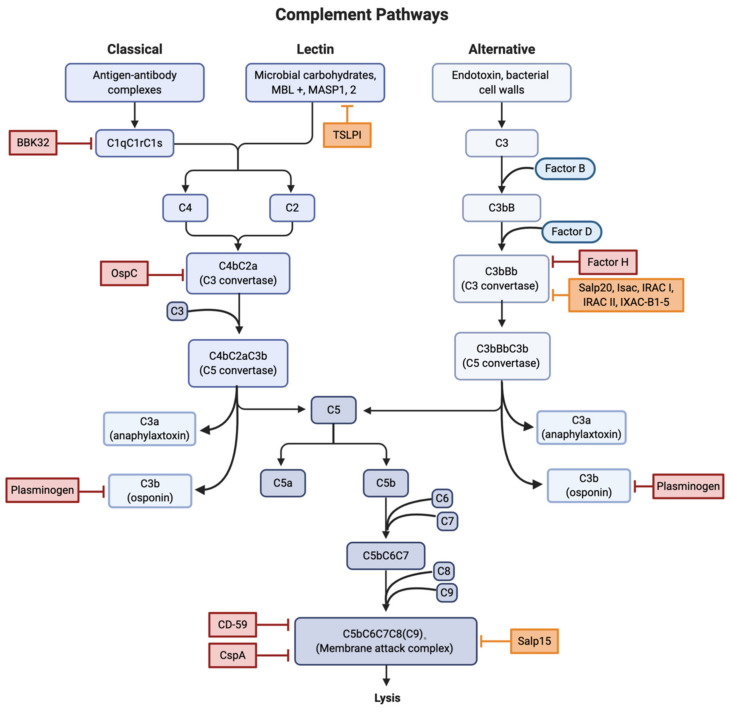
Locations of interference for all three complement activation pathways. Red boxes were used to indicate sites of both direct and indirect inhibition involved in *B. burgdorferi* infection. Orange boxes indicate sites of compliment inhibition due to tick salivary proteins. Abbreviations: Osp, outer surface protein; TSLPI, tick salivary lectin pathway inhibitor; Salp, salivary protein; Isac/Irac/Ixac, *Ixodes* anti-complement proteins. Created with BioRender.com.

**Figure 3 pathogens-10-00281-f003:**
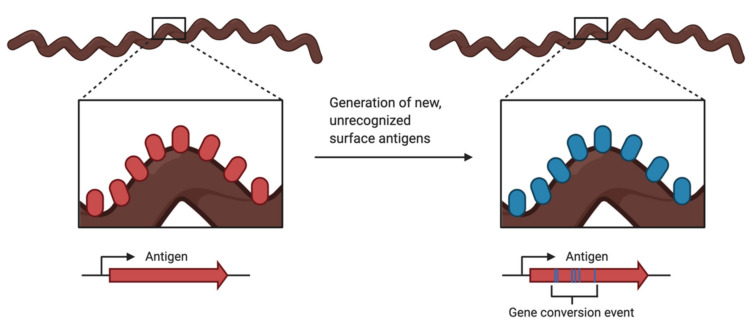
Illustration of antigenic variation, a tactic used by pathogens to avoid detection by the hosts immune system. Evasion is accomplished by the continuous change in a prominent surface antigen through gene conversion events or change in allelic expression. Created with BioRender.com.

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
