# Peer review of "The Brilliance of Borrelia: Mechanisms of Host Immune Evasion by Lyme Disease-Causing Spirochetes"

_pathogens, 2021, doi:10.3390/pathogens10030281_

Round 1

Reviewer 1 Report

Thoughtful, critical, and comprehensive review of how B.burgdorferi evades and impedes the host immune system.

The only suggestion is to add that there is clinical evidence to support the animal data that persistent IgM responses (along with few, if any IgG responses) are present in patients with persistent/chronic Lyme disease.

There are also indications that B.burgdorferi can evade and/or tolerate antibiotic treatments, the mechanisms to be further defined, but perhaps related to possible antibiotic-tolerant systems. Whether the authors wished to include discussion of these aspects, I leave up to them.

Author Response

Thoughtful, critical, and comprehensive review of how B.burgdorferi evades and impedes the host immune system.

We thank the reviewer for the kind words.

The only suggestion is to add that there is clinical evidence to support the animal data that persistent IgM responses (along with few, if any IgG responses) are present in patients with persistent/chronic Lyme disease.

We have added a statement to this effect and added additional references (Lines 540-542).

There are also indications that B.burgdorferi can evade and/or tolerate antibiotic treatments, the mechanisms to be further defined, but perhaps related to possible antibiotic-tolerant systems. Whether the authors wished to include discussion of these aspects, I leave up to them.

We have added a statement to this effect and added additional references (Lines 344-356).

Reviewer 2 Report

This is a well-written review highlighting many of the known mechanisms employed by Lyme disease spirochetes along with contributions of their tick vectors for immune evasion.  

Minor comments:

  1. In the opening paragraph, why not include other known vectors for B. burgdorferi sensu lato such as Ixodes ricinus? Inclusion of vectors other than I. scapularis and I. pacificus would make sense, since other vectors are discussed later in the review.
  2. The sentence on lines 101-103 is redundant. CspA was already described in the previous paragraph (lines 83-85).
  3. An additional sentence relating the how the 5µg of Salp20 used for inhibition of the complement system corresponds with concentrations of Salp20 in tick saliva would be helpful. 
  4. Line 224: Rather than destroy cysteines, ROS oxidize cysteines.
  5. Line 224: RNS include NO, dinitrogen trioxide (N2O3), nitrogen dioxide (NO2), and ONOO-.
  6. Lines 237-238: There is no experimental evidence that oxidation of membrane lipids change the function of membrane-associated proteins.
  7. Lines 244-253: B. burgdorferi has additional antioxidant defenses. SodA is the only one shown to contribute to mammalian infection, however, it remains unclear whether its role is to detoxify ROS produced by the innate immune response or to simply limit the toxicity of endogenously produced ROS by B. burgdorferi.
  8. Lines 254-255: Change RNIs to RNS.

Author Response

This is a well-written review highlighting many of the known mechanisms employed by Lyme disease spirochetes along with contributions of their tick vectors for immune evasion.  

We thank the reviewer for the kind words.

Minor comments:

In the opening paragraph, why not include other known vectors for B. burgdorferi sensu lato such as Ixodes ricinus? Inclusion of vectors other than I. scapularis and I. pacificus would make sense, since other vectors are discussed later in the review.

Added lines 27-29: “Ixodes ricinus and Ixodes persulcatus are the tick vectors for the Borrelia burgdorferi sensu lato (s.l.) complex in Europe and Asia, respectively [5].”

The sentence on lines 101-103 is redundant. CspA was already described in the previous paragraph (lines 83-85).

We have left this line in for readability and connection between paragraphs.

An additional sentence relating the how the 5µg of Salp20 used for inhibition of the complement system corresponds with concentrations of Salp20 in tick saliva would be helpful. 

We have added a reference for concentrations of proteins in tick saliva (lines 146-147).

Line 224: Rather than destroy cysteines, ROS oxidize cysteines.

We have edited this sentence accordingly (line 251).

Line 224: RNS include NO, dinitrogen trioxide (N2O3), nitrogen dioxide (NO2), and ONOO-.

We have added these RNS species (lines 250-251).

Lines 237-238: There is no experimental evidence that oxidation of membrane lipids change the function of membrane-associated proteins.

We have eliminated this statement.

Lines 244-253: B. burgdorferi has additional antioxidant defenses. SodA is the only one shown to contribute to mammalian infection, however, it remains unclear whether its role is to detoxify ROS produced by the innate immune response or to simply limit the toxicity of endogenously produced ROS by B. burgdorferi.

We have adapted this section with this clarification (lines 316-321).

Lines 254-255: Change RNIs to RNS.

We have changed this (line 328).

Reviewer 3 Report

Review of the MS No. pathogens-1105303 entitled "Borrelia befuddlement: mechanisms of host immune evasion by Lyme disease causing spirochetes"

The review article shows how Borrelia spirochetes avoid immune response using their own and ticks' proteins, pleomorphism, intracellular localization, and invasion of the lymph nodes. The article is very well written and presents results of the studies from the last two decades.

I have only two specific comments

  1. Lines 26-28: Why are only Borrelia vectors from North America mentioned here? What about Europe and Asia? Further in the text the authors mentioned Ixodes ricinus several times, but did not write it's vector of Borrelia in Europe.
  2. Figure 2 - please add information about abbreviations such as TSLPI, Osp or Sapl. These are only in the text.

Author Response

The review article shows how Borrelia spirochetes avoid immune response using their own and ticks' proteins, pleomorphism, intracellular localization, and invasion of the lymph nodes. The article is very well written and presents results of the studies from the last two decades.

We thank the reviewer for the kind words.

Lines 26-28: Why are only Borrelia vectors from North America mentioned here? What about Europe and Asia? Further in the text the authors mentioned Ixodes ricinus several times, but did not write it's vector of Borrelia in Europe.

We have added lines 27-29: “Ixodes ricinus and Ixodes persulcatus are the tick vectors for the Borrelia burgdorferi sensu lato (s.l.) complex in Europe and Asia, respectively [5].”

Figure 2 - please add information about abbreviations such as TSLPI, Osp or Sapl. These are only in the text.

We have added abbreviations to the figure legend (line 75-76).